# Primary care provider uptake of intensive behavioral therapy for obesity in Medicare patients, 2013–2019

**Mounira Ozoor**[1], **Mark Gritz**[2], **Rowena J. Dolor**[3], **Jodi Summers Holtrop**[4], **Zhehui Luo**[1]*

1 Department of Epidemiology and Biostatistics, Michigan State University, East Lansing, Michigan, United States of America, 2 Division of Health Care Policy and Research, Department of Medicine, University of Colorado School of Medicine, Aurora, Colorado, United States of America, 3 Division of General Internal Medicine, Department of Medicine, Duke University School of Medicine, Durham, North Carolina, United States of America, 4 Department of Family Medicine, University of Colorado School of Medicine, Aurora, Colorado, United States of America

* zluo@msu.edu

## Abstract

**Data Availability Statement:** Data are from the Center for Medicare and Medicaid Services, America's Health Ranking, and Kaiser Family Foundation. The URLs are: https://data.cms.gov/provider-summary-by-type-of-service/medicare-

### Background

The delivery of Intensive Behavioral Therapy (IBT) for obesity by primary care providers (PCPs) has been covered by Medicare to support elderly individuals with obesity (BMI > 30 kg/m$^2$) in managing their weight since 2011 for individual therapy and 2015 for group therapy. We conducted a cohort study of PCPs in an attempt to understand patterns of uptake of IBT for obesity services among PCPs serving the Medicare population across the U.S.

### Methods

We used the Centers for Medicare and Medicaid Services Provider Utilization and Payment Data from 2013 to 2019 to identify all PCPs who served more than 10 Medicare beneficiaries in each of the seven-year period to form a longitudinal panel. The procedure codes G0447 and G0473 were used to identify PCPs who provided IBT; and the characteristics of these providers were compared by the IBT-uptake status.

### Results

Of the 537,754 eligible PCPs who served Medicare patients in any of the seven years, only 1.2% were found to be IBT service providers in at least one year from 2013 through 2019 (246 always users, 1,358 early adopters, and 4,563 late adopters). IBT providers shared a few common characteristics: they were more likely to be male, internal medicine providers, saw a higher number of Medicare beneficiaries, and practiced in the South and Northeast regions. Having higher proportion of patients with hyperlipidemia was associated with higher likelihood of a provider being an IBT-user.

physician-other-practitioners/medicare-physician-
other-practitioners-by-provider-and-service/data/
https://data.cms.gov/provider-summary-by-type-
of-service/medicare-physician-other-practitioners/
medicare-physician-other-practitioners-by-
provider/data/ https://www.
americashealthrankings.org/explore/annual https://
www.kff.org/other/state-indicator/distribution-by-
age/?dataView=1¤tTimeframe=
7&selectedDistributions=65&sortModel=%7B%
22colId%22:%22Location%22,%22sort%22:%
22asc%22%7D.

**Funding:** The initials of the author who received the award: JSH The grant number awarded to the author: HS024843 The funder: Agency for Healthcare Research and Quality URL of funder website: https://www.ahrq.gov/funding/index.html The funders had no role in study design, data collection and analysis, decision to publish, or preparation of the manuscript.

**Competing interests:** The authors have declared that no competing interests exist.

**Abbreviations:** BMI, Body mass index; CMS, Centers for Medicare and Medicaid Services; HCPCS, Healthcare Common Procedure Coding System; IBT, Intensive behavioral therapy; PCP, Primary care provider.

## Conclusions

Very few PCPs continuously billed IBT services for Medicare patients with obesity. Further investigation is needed to mitigate barriers to the uptake of IBT services among PCPs.

## Introduction

The prevalence of obesity in the U.S. has been rising [1, 2] and is currently at an all-time high [3]. As of 2018, approximately 30% of the adult population and 28.5% of the senior population are living with obesity [4]. As they age, the baby-boomer generation has higher rates of obesity than previous generations [5]. Obesity is one of the major drivers of preventable health care costs, which is estimated between $147 billion and $210 billion [6]. In order to address this challenging public health issue, in 2011, the Centers for Medicare and Medicaid Services (CMS) established a Healthcare Common Procedure Coding System (HCPCS) code and authorized primary care providers (PCPs) to deliver and bill for Intensive Behavioral Therapy (IBT) for obesity in an attempt to help treat individuals with obesity (BMI $\geq$ 30 kg/m$^2$) [7].

IBT is an evidence-based service with a "B" rating, meaning there is "high certainty that the net benefit is moderate or there is moderate certainty that the net benefit is moderate to substantial," and is recommended by the United States Preventive Services Task Force to be provided by clinicians to eligible patients [8]. Some studies found that high intensity counseling, as well as behavioral interventions, will deliver continuous weight loss for individuals with obesity [9]. IBT consists of working closely with an approved provider to target behaviors that are contributing to a patient's obesity condition and educate the patient and implement changes such as tracking dietary intake and creating an exercise plan. All Medicare beneficiaries with obesity are eligible for the service [10]. The benefit allows for 15-min visits once per week for 4 weeks, biweekly visits for months 2 to 6, and then once monthly for another 6 months if the patient lost $\geq$ 3 kg (6.6 lb) [11]. To encourage Medicare beneficiaries to receive IBT services, CMS waived the Medicare coinsurance and Part-B deductible for this service. However, CMS established several requirements that must be met for PCPs to receive reimbursement for delivering IBT services to beneficiaries with obesity, including: (1) the service must be provided in a primary care setting by a qualified PCP; (2) a total of 22 visits can be billed in a 12-month period; (3) patients are seen once per week in the first month and twice per month through month 6; and (4) patients must meet a weight loss goal of at least 3 kg (6.6 lbs.) during the first 6 months to continue treatment, which is expected to be once a month for the next 6 months. If the patient does not achieve the weight loss goal, they must wait at least 6 months and be assessed for eligibility based on BMI in order to begin a new treatment period once again [11].

IBT services are reimbursed through two HCPCS codes. The first code, G0447, was authorized for services beginning in November 2011 and is used for one-on-one face-to-face behavioral counseling for obesity for a 15-minute encounter. The second code, G0473, was authorized beginning in January 2015 for face-to-face group (2–10 patients) behavioral counseling for obesity for a 30-minute group session. The average reimbursement rates for each service is $24-$26 for G0447 and $12-$13 for G0473 in 2018 [12].

The uptake of IBT services among Medicare providers has been underwhelming (0.1% of eligible beneficiaries in 2012 [13] and 0.2% in 2015 [14]) since the CMS authorized billing in 2011. The purpose of this paper is to examine the pattern of uptake of IBT for obesity services among PCPs serving the Medicare fee-for-service population across the U.S. to identify characteristics of PCPs who provided the services early, late, or never after the activation of the

codes. Our analysis of provider characteristics complements and adds to our understanding of the utilization of IBT services by beneficiary characteristics [13, 14].

## Methods

We conducted a cohort analysis of PCPs who were authorized to provide IBT services and who billed Medicare for more than ten unique beneficiaries for any service between 2013 and 2019 to examine their IBT service uptake patterns. The study was approved by the Institutional Review Boards of University of Colorado and Michigan State University as non-human subject research.

### Data sources

The publicly available Medicare Fee-for-service Provider Utilization and Payment data for Physician and Other Supplier Public Use File as well as the Provider Summary Tables from CMS covering calendar years 2013 through 2019 are used to identify the primary care providers eligible for delivering IBT [15]. The public use file includes providers who had a valid National Provider Identifier and submitted Medicare Part-B non-institutional claims in a calendar year. To protect the privacy of Medicare beneficiaries, any aggregated records that are derived from 10 or fewer beneficiaries are excluded from the public use data, which led to some providers being classified as a non-user even though they had used IBT for a handful of patients. The data elements include the provider's identifier, provider type, gender, zip code, state, HCPCS codes billed for more than 10 unique beneficiaries by the provider, and number of claims, unique beneficiaries and average Medicare payment amount by HCPCS codes. The Provider Summary Tables include data pertaining to service utilization, payments, provider demographics, and beneficiary demographics. America's Health Ranking's obesity data [16] and Kaiser Family Foundation's publications [17] are used to find Medicare beneficiary populations with obesity by state.

### Data analysis

We restricted the PCP types to family practice, general practice, internal medicine, nurse practitioner and physician assistant as these were the PCPs who were authorized to bill under the two IBT HCPCS codes. If a provider who was included in the analysis did not have any of the two HCPCS codes in a given year, the provider was considered an *IBT non-user* for that year. We categorized the providers based on their IBT-uptake pattern over this period: *always users* include providers who had IBT claims for more than 10 unique Medicare beneficiaries every year during 2013 through 2019; *early adopters* include providers who had at least some utilization during 2013 or 2014 but not always in the other five years; *late adopters* include providers who did not have IBT claims reported during 2013 and 2014, but did during at least one year in 2015 through 2019, and lastly, *never users* include providers who had no reported payments for IBT claims from 2013 through 2019. Within each of these categories, we examined the following provider characteristics: gender, region, provider type, average count of unique Medicare beneficiaries, provider charges and beneficiaries with other relevant chronic conditions served by the provider.

We examined the IBT service penetration in two ways: the number of Medicare beneficiaries per 1,000 elderly population with obesity in a state, and the number of PCPs who billed for IBT per 1,000 PCPs in a state. Choropleth maps were used to show utilization patterns between 2013 and 2019 within the measurement period across states. These maps can identify regions of strong and weak adoption of the services.

Descriptive statistics (sample size, proportions for discrete variables, medians, means and standard deviations for continuous variables) were presented by provider user types.

Multinomial logistic regressions were used to compare provider characteristics and patient-compositions related to utilization patterns. Two-sided tests at the 0.05 significance level were performed using Stata 17 (StataCorp LLC).

## Results

Table 1 shows provider characteristics by their IBT-uptake patterns (*never users*, *late adopters*, *early adopters* and *always users*) for the 537,754 providers identified in the public use data. Of these providers, the majority of them were considered *never users* (98.9%); of the *never users*, 61.1% were females, mainly practicing in the South (36.3%) and Midwest (24.1%), and primarily made up of internal medicine providers (24.8%), nurse practitioners (34.0%), and family practice providers (20.2%). The *never users* had a median count of 108 unique Medicare beneficiaries per year when considering all Medicare claims.

**Table 1. Provider characteristics by intensive behavioral therapy (IBT) uptake patterns (N = 537,754) from 2013 to 2019.**

| | IBT Providers* | | | | | | | |
| --- | --- | --- | --- | --- | --- | --- | --- | --- |
| | Never Users (n = 531,587) | | Late Adopters (n = 4,563) | | Early Adopters (n = 1,358) | | Always Users (n = 246) | |
| | n | % | n | % | n | % | n | % |
| Gender | | | | | | | | |
| Males | 206,059 | 38.9% | 1,975 | 56.7% | 831 | 61.2% | 167 | 67.9% |
| Females | 325,059 | 61.1% | 2,588 | 43.3% | 527 | 38.8% | 79 | 32.1% |
| Region: | | | | | | | | |
| Midwest | 128,375 | 24.1% | 696 | 15.3% | 188 | 13.8% | 22 | 8.9% |
| Northeast | 109,266 | 20.6% | 1,150 | 25.2% | 347 | 25.6% | 90 | 36.6% |
| South | 193,094 | 36.3% | 1,951 | 42.5% | 606 | 44.6% | 104 | 42.3% |
| West | 100,717 | 18.9% | 776 | 17.0% | 217 | 16.0% | 30 | 12.2% |
| Provider Type: | | | | | | | | |
| Family practice | 107,270 | 20.2% | 1,692 | 37.1% | 485 | 35.7% | 78 | 31.7% |
| General practice † | 7,428 | 1.4% | 106 | 2.3% | 35 | 2.6% | 3 | 1.2% |
| Internal medicine | 132,094 | 24.8% | 1,951 | 42.8% | 713 | 52.5% | 161 | 65.4% |
| Nurse practitioner | 180,882 | 34.0% | 611 | 13.4% | 93 | 6.8% | 2 | 0.8% |
| Physician assistant | 103,913 | 19.5% | 203 | 4.4% | 32 | 2.4% | 2 | 0.8% |
| Median annual number of unique Medicare beneficiaries | 108 | | 266 | | 329 | | 350 | |
| Median annual number of services/encounters per year | 264 | | 1,624 | | 2,804 | | 4,202 | |
| Median annual number of all procedure codes | 16 | | 42 | | 55 | | 65 | |
| Median annual submitted charge amount | $43,955 | | $166,638 | | $273,253 | | $354,981 | |
| Percent of beneficiaries with obesity related chronic conditions ‡ | | | | | | | | |
| Hypertension | 68.0% | | 69.6% | | 70.6% | | 71.7% | |
| Diabetes | 38.2% | | 38.8% | | 39.0% | | 42.9% | |
| Hyperlipidemia | 57.1% | | 60.9% | | 60.9% | | 66.1% | |

* Always users include providers who provided IBT services to more than 10 Medicare beneficiaries every year during 2013 through 2019; early adopters include providers who had provided IBT services to more than 10 Medicare beneficiaries per year during 2013 or 2014 but not in the other five years; late adopters include providers who had never provided IBT services to more than 10 Medicare beneficiaries per year during 2013 and 2014, but had done so at least one year from 2015 through 2019; and never users include providers who had never provided IBT services to more than 10 Medicare beneficiaries during 2013 through 2019.

† The provider type "general practice" is used by the Medicare Fee-for-service Provider Utilization and Payment data for Physician and Other Supplier Public Use File as a separate category, as are nurse practitioner and physician assistant.

‡ Percentages based on providers with available data from Provider Summary Tables.

*Late adopters* comprised just 4,563 (0.9%) providers. Most of these providers were males (56.7%), practicing mainly in the South (42.5%) and Northeast (25.2%). The *late adopters* were mainly internal medicine (42.8%) and family practice (37.1%) providers, with a median count of 266 unique Medicare beneficiaries per year.

*Early adopters* consisted of only 1,358 (0.3%) providers, with 61.2% being males. The *early adopters* mainly practiced in the South (44.6%) and the Northeast (25.6%) and were primarily internal medicine (52.5%) and family practice (35.7%) providers. The early adopters had a median count of 329 unique Medicare beneficiaries per year.

Lastly, the smallest group, the *always users*, consisted of only 246 providers (0.05%), who shared a similar distribution of characteristics as the early and late adopters. These providers were 67.9% male, practicing mainly in the South (42.3%) and Northeast (36.6%), and made up of mostly internal medicine (65.4%) and family practice (31.7%) providers. The *always users* had a median count of 350 unique Medicare beneficiaries per year.

We also utilized the Provider Summary Tables to help further understand the distributions of several other variables among the different IBT-uptake patterns, including the median numbers of total annual services or encounters, annual unique billed HCPCS codes, and annual total submitted charges. Generally the services rendered were lowest among non-IBT providers and highest among the *always users* (Table 1).

We also examined the distributions of obesity related chronic conditions, including each provider's average proportions of Medicare patients with hypertension, diabetes, and hyperlipidemia by the IBT-uptake patterns. The proportions of patients with these chronic conditions were fairly similar but did slightly increase across the four IBT-uptake groups (lower panel Table 1), showing somewhat of a pattern between the prevalence of the relative chronic conditions in the populations that the providers served and whether or not they provided IBT obesity services to their patients.

Table 2 showed the odds ratio (OR) of a multinomial logistic regression with the four uptake patterns as the outcome and factors in Table 1 as regressors (excluding the highly

**Table 2. Multinomial logistic regression of IBT update patterns and provider characteristics and patient composition.**

| | Never Users | Late Adopters | | Early Adopters | | Always Users | |
|---|---|---|---|---|---|---|---|
| | | OR (95% CI) | | OR (95% CI) | | OR (95% CI) | |
| Male | 38.9% | Ref | | Ref | | Ref | |
| Female | 61.1% | 0.66 | (0.62, 0.71) | 0.64 | (0.57, 0.72) | 0.55 | (0.42, 0.74) |
| Midwest | 24.1% | Ref | | Ref | | Ref | |
| Northeast | 20.6% | 1.76 | (1.60, 1.93) | 1.82 | (1.52, 2.18) | 3.54 | (2.22, 5.66) |
| South | 36.3% | 1.67 | (1.53, 1.82) | 1.83 | (1.55, 2.16) | 2.49 | (1.57, 3.95) |
| West | 18.9% | 1.46 | (1.32, 1.62) | 1.56 | (1.28, 1.90) | 1.89 | (1.09, 3.28) |
| Other PCPs† | 75.2% | Ref | | Ref | | Ref | |
| Internal medicine | 24.8% | 1.49 | (1.40, 1.60) | 2.03 | (1.80, 2.28) | 2.84 | (2.12, 3.79) |
| # Medicare beneficiaries /100 | 1.8 | 1.08 | (1.07, 1.09) | 1.08 | (1.07, 1.09) | 1.08 | (1.06, 1.09) |
| # services encounters /100 | 9.4 | 1.00 | (1.00, 1.00) | 1.00 | (1.00, 1.00) | 1.00 | (1.00, 1.00) |
| $ submitted charges /10,000 | 11.2 | 1.04 | (1.03, 1.04) | 1.04 | (1.04, 1.05) | 1.05 | (1.04, 1.06) |
| % patients w/ hypertension /10 ‡ | 6.8 | 0.92 | (0.90, 0.96) | 0.95 | (0.86, 1.04) | 0.79 | (0.63, 1.17) |
| % patients w/ diabetes /10 ‡ | 3.8 | 0.93 | (0.90, 0.96) | 0.87 | (0.82, 0.91) | 1.05 | (0.94, 1.17) |
| % patients w/ hyperlipidemia patients /10 ‡ | 5.7 | 1.28 | (1.24, 1.32) | 1.50 | (1.41, 1.60) | 1.92 | (1.63, 2.26) |

† Other PCPs include family practice, general practice, nurse practitioner, and physician assistant.

‡ Missing data were imputed by the mean value. The percentages of missing data for hypertension, diabetes, and hyperlipidemia were 6.7%, 18.7%, and 11.2%.

collinear number of annual billed HCPCS codes). The provider types other than internal medicine were grouped together to avoid the small cell problem. The 95% confidence intervals (CIs) indicated that male, Northeast-, South- and West-region, and internal medicine providers had an increasing likelihood of being a late, early, or always adopters compared to never users. Compared with never users, for every 10 percentage points increase in patients with hypertension among later adopters, the OR for being a late adopter was 0.92 (95% CI 0.90, 0.96). Similarly, for every 10 percentage points increases in patients with diabetes the OR for being a late adopter was 0.93 (95% CI 0.90, 0.96) and the OR for being an early adopter was 0.87 (95% CI 0.82, 0.91). However, for every 10 percentage points increases in patients with hyperlipidemia, the OR for being a late adopter was 1.28 (95% CI 1.24, 1.32), the OR for being an early adopter was 1.50 (95% CI 1.41, 1.60), and the OR for being an always user was 1.92 (95% CI 1.63, 2.26). An alternative classification of providers showed similar results (see Supplemental materials).

Fig 1 displayed the total number of claim counts for IBT services and the total number of unique Medicare beneficiaries reported in the public use data from 2013 to 2019 for providers who delivered IBT services to more than 10 fee-for-service Medicare beneficiaries in a year. Although over the years there had been a steady increase in utilization, both in the number of claims and number of beneficiaries, only about 1% of the eligible Medicare population who met the criteria ever received IBT services from 2013 through 2019.

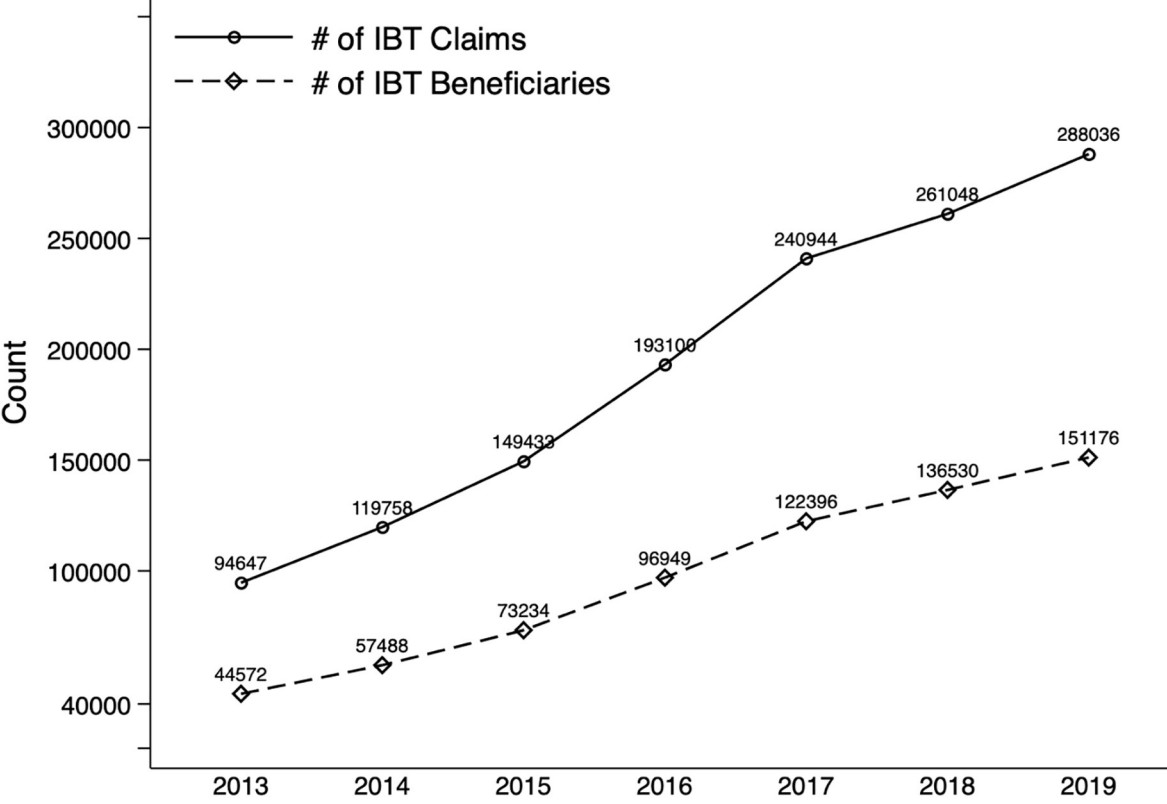

**Fig 1. Intensive behavioral therapy (IBT) for obesity services utilization patterns (2013–2019).** The number of claims for intensive behavioral therapy (IBT) for obesity services (solid line) and the number of beneficiaries who received the IBT services (dash line) documented in 2013 to 2019 Medicare Fee-for-Service Provider Utilization and Payment Data for Physicians and Other Suppliers.

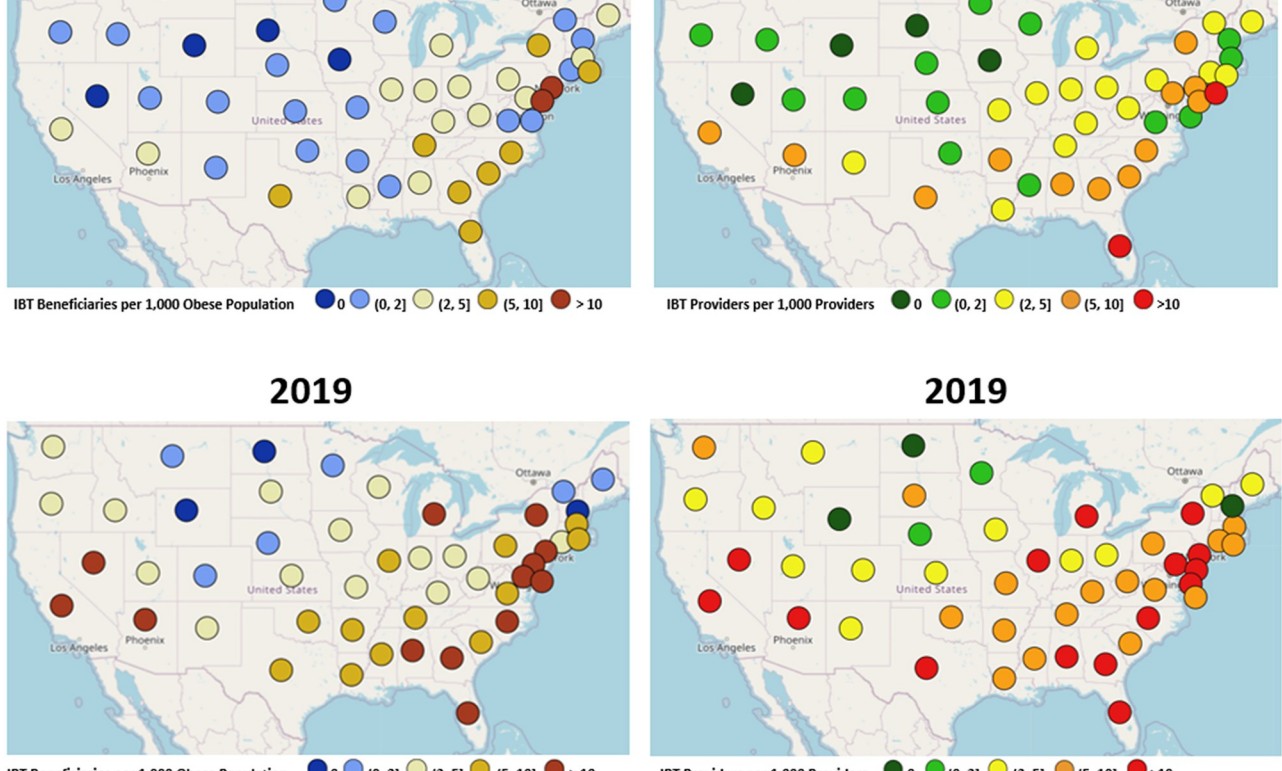

**Fig 2. Choropleth maps for intensive behavioral therapy (IBT) utilization by Medicare beneficiaries and primary care providers.** The number of Intensive behavioral therapy (IBT) beneficiaries per 1,000 Medicare population with Obesity (left panel) and the number of providers per 1,000 eligible primary care providers (right panel) who provided IBT services to more than 10 Medicare beneficiaries across the United States in 2013 and 2019.

Fig 2 presented choropleth maps of the number of Medicare beneficiaries who received IBT services per estimated 1,000 Medicare beneficiaries with obesity in each state for 2013 and 2019 (left panel), alongside the number of IBT providers delivering IBT services to more than 10 beneficiaries in a year per 1,000 eligible providers (right panel). Due to copyright issues, we had to use an open source tool at USGC.gov to recreate these maps which lost the formatting option of the states having filled colors that represented their values and only allowed us to manually select and plot different colored markers to represent the state's value (For alternative format see S2 File). There was a faster increase in IBT beneficiaries occurring mainly in the South overall, however two states, New Jersey and New York, in the Northeast stood out and had the highest rate of IBT beneficiaries per 1,000 beneficiaries with obesity overall in 2019 (27.5 and 26.4, respectively). Although most providers were classified as *never users*, the choropleth maps confirmed the regional pattern that IBT uptake increased primarily in the South over the study time period, but once again New Jersey and New York had the overall highest rate of IBT Providers per 1,000 providers in 2019 (22.5 and 19.3, respectively). The state that saw the largest increase in the absolute number of IBT providers per 1,000 providers was Nevada (0 to 18.3 from 2013 to 2019); and New Hampshire and Vermont were the only two states that had a slight overall decrease in the number of IBT providers per 1,000 providers from 2013 to 2019 (1.0 to 0.0 and 2.4 to 2.0, respectively).

## Discussion

This study was the first to classify providers by the IBT uptake timing to early, late, always or never adopter groups and examine differences between these groups. There was a pattern in the characteristics of providers delivering IBT services to more than 10 Medicare beneficiaries in a calendar year. Specifically, being female, non-internal medicine PCP, practicing in the Midwest region and small clinics were associated with lower probabilities of being an IBT user. Having higher proportions of patients with hyperlipidemia was associated with higher probabilities of being a late adopter, an early adopter, or an always user. This relationship between the patient composition and provider IBT-uptake was consistent with the need for obesity management in patients with high lipids. These patterns were consistent with practice-level characteristics associated with IBT uptake (e.g., larger practices are more likely to use IBT) [18], geographic distribution of the Medicare population with obesity (e.g., prevalence of beneficiaries with obesity is highest in the South) [19], and the complexity of obesity management (e.g., some PCPs such as nurse practitioners need more formal training for treating obesity) [20].

Although IBT utilization in the Medicare population across the U.S. has been steadily increasing over the years since the implementation of the CMS procedure codes, it is still a highly underutilized benefit among the Medicare population who qualify for the service. Approximately 28.5% of Medicare beneficiaries are estimated to meet criteria for obesity, suggesting there are about 10 to 11 million fee-for-service beneficiaries that meet the criteria to receive IBT services per year [4, 21]. There are likely many factors affecting providers' and Medicare beneficiaries' decisions that result in the very low utilization of IBT for obesity treatment. Among the factors that may affect providers' decisions to deliver IBT for obesity treatment are some of the structural features of the billing and payment requirements CMS established for the two HCPCS codes for individual and group sessions. For example, one potentially important factor is the low reimbursement rates for both individual and group sessions relative to primary care evaluation and management reimbursement levels. The rate of reimbursement for a 15-min consultation of IBT is $26, whereas payment for a typical evaluation and management code for an established patient for a level-2 visit (CPT code 99212) is $45 and a level-3 visit is $74 (CPT code 99213, Medicare payment for calendar year 2018). While CMS aimed to increase utilization of preventive service and screenings through procedure codes, the uptake rates of these services have been low. For example, 3% of Medicare FFS beneficiaries received a visit specifically addressing depression screening in 2016 [22]. The G-codes for Annual Wellness Visits were introduced in 2011 with much higher allowable charge than obesity counseling services but the penetration was only 16~17% in 2014 [23, 24].

Another factor is likely the expectation of weekly visits for the first month and twice a month for the next five months, which for busy primary care practices could significantly curtail the available appointments for their entire patient panel. In addition, these frequent visits may be difficult for patients to achieve in certain populations with other comorbid conditions such as diabetes and patients with more resource constraints such as transportation barriers.

Another important barrier to uptake is the restriction that the service needs to be provided in a primary care setting by a physician, nurse practitioner, physician assistant or a qualified provider under their direct supervision [25]. A non-physician auxiliary practitioner, such as a registered dietician (RD), may provide IBT but they must bill "incident to" the primary care physician, who must be physically present at the time services are provided [11]. Referral to RDs and other auxiliary providers who work outside of the primary care setting is not covered by CMS [11]. In the Medicare Fee-for-service data from 2013 to 2019, we found the following provider specialty types who submitted Medicare claims for IBT services: general practice,

family practice, internal medicine, obstetrics/gynecology, pediatric medicine, geriatric medicine, nurse practitioner, certified nurse specialist, or physician assistant. Had other providers been certified to provide obesity service it might increase uptake of the benefit. In Table 3, we summarized a list of barriers as well as potential solutions for low IBT provider-uptake and patient-utilization rates based on our findings from provider interviews and surveys [18, 20].

Two similar studies have examined the uptake rates of IBT services since its implementation in 2012 [13, 14]. The first study examined utilization in 2012 and 2013 [13] and the second study for 2012 through 2015 [14]. These two studies showed very similar results in terms of overall low IBT utilization. However, in contrast to our study, their analyses focused on Medicare beneficiary characteristics and not provider characteristics. Using patient-level claims data, both studies described the demographics of patients who received IBT services. Our study focused on provider characteristics by IBT-uptake status and geographic differences, which have not yet been evaluated in similar studies. This new information allows us to further understand where IBT uptake is extremely low and how it varies across different provider types. Also, these results examine the data over a longer period of time, indicating the trend towards low utilization has continued.

At the time of the study, the 2020 data became available. However, we decided to exclude it from our analyses because the data may be misleading due to the effect on health care visits of the early part of the COVID-19 pandemic. The number of IBT claims dropped to 245,442 (a 15% drop compared with 2019 data) and the number of unique beneficiaries dropped to 134,560 (a 11% drop). Access to non-emergent health care services was limited during the last months of 2020.

One limitation of this work is that we were not able to assess providers who did provide IBT for obesity but only for 10 or fewer beneficiaries because of data suppression in the CMS publicly available data, which may lead to a slight underestimation. Some late adopters may have been misclassified when they used some IBT services in the early years to a lesser extent. Compared with the previous report [14] our estimated number of beneficiaries in 2013 was slightly lower (46,821 from the previous report versus our finding of 44,572). Therefore, our results may underrepresent the true amount of IBT for obesity happening in the earlier years. However, our estimates in 2014 (46,171 from the previous report versus our finding of 57,488) and 2015 (57,576 from the previous report versus our finding of 73,234) were higher which might be due to the updates of final-action claim items over time. The differences in our estimates and previous reports were small and would not change the qualitative conclusions of the study.

**Table 3. Barriers and potential solutions for low IBT provider uptake and patient utilization rates.**

| Barriers to IBT service provider uptake and patient utilization | Potential Solution |
| --- | --- |
| Reimbursement does not fully cover cost of clinician personnel, facility, and program (care coordination) expenses. | Add coverage for care coordination and facility expenses. |
| Requirement for delivery under direct supervision of PCP. | Broaden the types of providers (registered dieticians, behavioral health workers, nurses, social workers) who can deliver IBT and bill independently. |
| Requirement for weekly (first month) and biweekly (month 2 and beyond) in-person visits is burdensome to patients. | Allow telehealth delivery of IBT services. |
| Requirement for specified weight loss of 3 kgs in the first 6 months to continue IBT billing. | Remove weight loss specification. Behavioral changes require time to take effect. Obesity is a chronic condition, not subacute or temporary one. |

Another limitation of the public use data is that it only contains information on Medicare fee-for-service beneficiaries and as such it may not be representative of a physician's entire practice pattern. A third limitation of the study is that when we classified providers into early, late, always or never adopters we did not exclude providers who were not in the National Plan and Provider Enumeration System until after 2014 or providers who were deactivated between 2013 and 2019. The number of providers in the cohort would be smaller if we had done so. We found 141,313 out of the 575,936 PCPs in our cohort had the first enumeration year after 2014, among whom 140,808 were classified as never users in our analysis and 505 as late adopters. Using the CMS Deactivated Providers list as of July 2022, we found 6,049 of the 575,936 PCPs in our cohort were deactivated in the study period, among whom 5,985 were never users, 35 were late adopters and 29 were early adopters. We do not expect these limitations to affect our results qualitatively.

Our study provides the targeting regions and provider types to increase utilization in the future. Many structural changes may be needed to improve the uptake of IBT services for Medicare beneficiaries. Low reimbursement rates relative to other services provided by or under the direct supervision of PCPs are often cited as a reason PCPs do not offer IBT or similar types of services to Medicare beneficiaries [26]. Another structural change CMS could adopt to increase beneficiary access to IBT for obesity services is to remove the requirement for delivery by or under the direct supervision of a PCP. Registered dietitians are authorized Medicare Part B providers and CMS could change the coverage determination to permit them to independently deliver and bill for IBT for obesity services. As evidenced by the recent rapid transition to more telehealth services, IBT for obesity services are a good candidate for consideration of delivery via telehealth modes that would enhance the access to these services by Medicare beneficiaries who face transportation barriers in accessing in-person services. Finally, these results identify states with higher uptake that researchers can target for a further study regarding practice characteristics that are related to successful adoption of the service.

## Conclusion

This study documents the extremely low use of IBT for obesity benefit in Medicare and regional variations in uptake rates with the South and Northeast U.S. regions having the highest number of IBT-billing providers. Adopters of the IBT services tend to differ systematically from the never users. Without CMS actively improving the benefit design this valuable service will continue to be underutilized. Further investigation is needed to identify barriers to the uptake of IBT services among PCPs.

## Supporting information

**S1 File. Supplemental materials.**
(DOCX)

**S2 File. SAS codes for generating heatmaps "IBT Heat Maps".**
(SAS)

## Author Contributions

**Conceptualization:** Mounira Ozoor, Mark Gritz, Rowena J. Dolor, Jodi Summers Holtrop, Zhehui Luo.

**Data curation:** Mounira Ozoor, Zhehui Luo.

**Formal analysis:** Mounira Ozoor, Zhehui Luo.

**Funding acquisition:** Jodi Summers Holtrop.

**Methodology:** Mounira Ozoor, Mark Gritz, Rowena J. Dolor, Jodi Summers Holtrop, Zhehui Luo.

**Supervision:** Zhehui Luo.

**Validation:** Mounira Ozoor, Zhehui Luo.

**Writing – original draft:** Mounira Ozoor, Zhehui Luo.

**Writing – review & editing:** Mounira Ozoor, Mark Gritz, Rowena J. Dolor, Jodi Summers Holtrop, Zhehui Luo.

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
