## [Decision Letter · Decision Letter 0]

7 Jul 2022

PONE-D-22-07772Primary Care Provider Uptake of Intensive Behavioral Therapy for Obesity in Medicare Patients, 2013-2019PLOS ONE

Dear Dr. Luo,

Thank you for submitting your manuscript to PLOS ONE. After careful consideration, we feel that it has merit but does not fully meet PLOS ONE’s publication criteria as it currently stands. Therefore, we invite you to submit a revised version of the manuscript that addresses the points raised during the review process.

ACADEMIC EDITOR:Dear Autor: Thank you for your submission. Before we can proceed any further, reviewers have pointed out important issues that should be address. Remember Aim/Objective should be sound stated scientifically sounded and conclude accordingly: This:  "The purpose of this paper is to improve our understanding of the underutilization as well as the pattern of utilization of IBT for obesity services among PCPs serving the Medicare population across the U.S" does not give a clear idea of what is expected to be done  Please also provide de STROBE Checklist for your submission Best regards==============================

We look forward to receiving your revised manuscript.

Kind regards,

Juan A López-Rodríguez

Academic Editor

PLOS ONE

Journal Requirements:

2. ‘Please include your tables as part of your main manuscript and remove the individual files. Please note that supplementary tables (should remain/ be uploaded) as separate "supporting information" files.’

4. We note that Figure (2) in your submission contain copyrighted images. All PLOS content is published under the Creative Commons Attribution License (CC BY 4.0), which means that the manuscript, images, and Supporting Information files will be freely available online, and any third party is permitted to access, download, copy, distribute, and use these materials in any way, even commercially, with proper attribution. For more information, see our copyright guidelines: http://journals.plos.org/plosone/s/licenses-and-copyright.

1. You may seek permission from the original copyright holder of Figure (2) to publish the content specifically under the CC BY 4.0 license. 

Reviewers' comments:

Reviewer's Responses to Questions

**Comments to the Author**

1. Is the manuscript technically sound, and do the data support the conclusions?

Reviewer #1: Partly

Reviewer #2: Partly

2. Has the statistical analysis been performed appropriately and rigorously? 

Reviewer #1: I Don't Know

Reviewer #2: No

3. Have the authors made all data underlying the findings in their manuscript fully available?

Reviewer #1: Yes

Reviewer #2: Yes

4. Is the manuscript presented in an intelligible fashion and written in standard English?

Reviewer #1: Yes

Reviewer #2: Yes

5. Review Comments to the Author

Reviewer #1: The manuscript reports a description of data from publicly available sources, about use and implementation of a funded service directed to Medicare-beneficiaries with obesity. The study ads a new piece to the puzzle of understanding why the use of this service is low. Below I provide a few more general concerns with the analyses and how the manuscript is written, as well as minor comments to the text.

GENERAL CONCERNS

First, the tables referred to in the text were not available in the files I was sent (tables 1 and 2).

Can you provide a more general discussion about how to interpret and understand the findings from this study outside of the fact that implementation is low? You found characteristics that identified service providers more or less likely to provide the service, but I can’t see any discussion about why the distribution of characteristics looks as it does or how this can be used to improve its use among those who are already eligible to provide it. Can you for example explain why female providers were less likely to take up this service, or what it is in their practices that makes uptake lower? In the conclusions you say that adopters differ systematically, but I can’t read out from your paper anything about the why, are for example some providers more exposed to the barriers you have identified?

Look through the methods section, in particular the Data analysis-section, and make sure all conducted analyses are presented here. It is now focusing on the data, not the analyses presented in the Results section. Moreover, I believe your paper would benefit from adding some more statistical analyses, such as comparing proportions between groups and variation in the number of beneficiaries per provider (mean and confidence intervals for example). Even regressions to identify to joint effect of different characteristics on use could be considered. Now it’s very descriptive, just really presenting the aggregated data without in depth analysis.

I believe readers would have use of a more thorough description of the intervention/service you are studying. Of course, its possible to go to other sources for details, but now its only very briefly described. Who is this service for, who is eligible, what does it include, how is it provided?

MINOR COMMENTS

Overall, the text suffers from many abbreviations. Would it be possible to reduce?

Background:

For me it was a bit unclear what “B” rating and B deductible is, and maybe for many other non-US readers.

What do you mean by “reassessed”, I could not see from the text that beneficiaries were ever assessed in the first place?

You say the uptake “has been underwhelming”, but how bad and is there a reference for that?

Methods:

So seldom-users are excluded, I guess this means that it is possible that some provider characteristics in the groups had changed if you included all users, and that e.g., some late adopters could even be always users? Now you use an estimate of 5000 seldom users, but without reference or argument for it. Can this be clarified in the methods and also a bit more discussed in the limitations section? Even an analysis showing how use is distributed by characteristic could provide some more information to this issue, is there for example provider groups that are very close to being listed as non-users because of this limit of 10 beneficiaries?

What is the study design, should I view this as an ecological study or how linked are the data sources?

Results:

Could you discuss your results that chronic conditions were similar regardless IBT-uptake, how should I interpret this?

Just to be clear I’m reading this correct, the numbers in parentheses that reports the number of beneficiaries per 1000 patients with obesity are e.g., 27.5 of 1000, and not percentages, so even in the areas with highest uptake its less than 3% of beneficiaries?

Discussion:

You write about low reimbursement rates. How low is it, compared to alternative uses of the same amount of time by these providers? If this is not prioritized, what is and why?

What do you mean by “these requirements may be difficult for patients”, what is requirements here?

What do you mean by “under their direct supervision”? Would it for example be enough to have a dietician be employed by one of the eligible providers?

You present numbers within parentheses (e.g., “46,821 versus 44,572”) but its difficult to see which number is which, could you add years in the parentheses also?

How many providers are expected to be among the “not in the National plan” until after 2014, or deactivated during the study period?

Reviewer #2: The manuscript presents updated data on the usage of the CMS IBT benefit by primary care providers between 2013 and 2019. Updated data and information about trends over time in the adoption of the CMS IBT benefit are useful and the inclusion of provider characteristics provides novel information not included in previous reports of CMS IBT uptake. I would recommend revisions that would make the adopter subgroups more useful to the reader as well as allow more definitive conclusions about the differences between provider subgroups.

• Please use person first language throughout the manuscript.

• The tables are missing from the manuscript.

• I found the early adopters vs late adopter categories to be less useful because they do not provide much information about the frequency with which providers that only occasionally reached the threshold of 10 claims per year were using the IBT codes. For example, some of the “early adopters” might have continued to use IBT services for at least 10 patients in most years, while others may have tapered off completely. Other data provided in the Figures already captures more meaningful information about changes in provider usage over time. Perhaps a different division such as “rare users” who had 10 claims in <50% of years vs “frequent users” >50% of years would provide more meaningful information about usage patterns. The “always user” category could be maintained though it is quite small. Alternatively, the authors could simply present data showing the percent of providers who met the threshold of 10 beneficiaries in only 1 yr, in 2-3 of the yrs, 4-5 yrs, etc.

• Although I cannot fully review this information without the tables, it appears likely that the authors are making claims about differences between provider usage groups when no statistical comparisons were conducted.

6. PLOS authors have the option to publish the peer review history of their article (what does this mean?). If published, this will include your full peer review and any attached files.

Reviewer #1: **Yes: **Hanna Gyllensten

Reviewer #2: **Yes: **Jena S. Tronieri

---

## [Author Response · Author response to Decision Letter 0]

12 Aug 2022

Please see the attached Response to Reviewers document.

---

## [Decision Letter · Decision Letter 1]

12 Sep 2022

PONE-D-22-07772R1Primary Care Provider Uptake of Intensive Behavioral Therapy for Obesity in Medicare Patients, 2013-2020PLOS ONE

Dear Dr. Luo,

Thank you for submitting your manuscript to PLOS ONE. After careful consideration, we feel that it has merit but does not fully meet PLOS ONE’s publication criteria as it currently stands. Therefore, we invite you to submit a revised version of the manuscript that addresses the points raised during the review process.

ACADEMIC EDITOR:  Dear Author Thank you very much for this quick answerOur reviewers ended the second review and still some minor changes are thought to be addressed. Please pay attention to those from reviewer 2 and some of the language, rewording and rephrasing recommendations. 

We look forward to receiving your revised manuscript.

Kind regards,

**Juan A López-Rodríguez**

**
*Academic Editor*
**

PLOS ONE

Journal Requirements:

Reviewers' comments:

Reviewer's Responses to Questions

**Comments to the Author**

1. If the authors have adequately addressed your comments raised in a previous round of review and you feel that this manuscript is now acceptable for publication, you may indicate that here to bypass the “Comments to the Author” section, enter your conflict of interest statement in the “Confidential to Editor” section, and submit your "Accept" recommendation.

Reviewer #1: All comments have been addressed

Reviewer #2: (No Response)

2. Is the manuscript technically sound, and do the data support the conclusions?

Reviewer #1: Yes

Reviewer #2: Yes

3. Has the statistical analysis been performed appropriately and rigorously? 

Reviewer #1: Yes

Reviewer #2: Yes

4. Have the authors made all data underlying the findings in their manuscript fully available?

Reviewer #1: Yes

Reviewer #2: Yes

5. Is the manuscript presented in an intelligible fashion and written in standard English?

Reviewer #1: Yes

Reviewer #2: Yes

6. Review Comments to the Author

Reviewer #1: Thank you for your careful revisions. I have no further comments and also learnt a couple of things from reading it, both about the Medicare services and the distinction between cohort analysis (a term I must admit I had not heard before, despite my background in register-based research) and cohort study.

Reviewer #2: I thank the authors for their revisions, which have improved and clarified the manuscript.

Person first language refers to separating the person from words that describe a medical or mental health condition of that person. This would mean changing all instances of “obese people/individuals/beneficiaries etc.” to “people/individuals/beneficiaries with obesity” and changing statements like “x% of the population are obese” to “x% of the population have obesity”. This format is now a standard requirement in most research publications. Please modify your language throughout the abstract and manuscript; they do not currently use person first language. I would also avoid using terms like “poor habits” which imply a value judgment and instead just use “habits (or behaviors) that contribute to the patient’s obesity (p. 3 of track changes text).

I actually would recommend against the re-calculation of the usage pattern groups based on the added 2020 data. The inclusion of 2020 further reduced the size of the “always user” group. As the authors note regarding their decision not to graph utilization for 2020 (p. 6), the utilization in that year was likely lower than previous years due to the pandemic. The IBT benefit requires patients to attend in person visits, and it is unlikely that many medical providers would have recommended that patients with obesity (who were at high risk of COVID-19 complications) attend in-person elective counseling early in the pandemic. There also were likely regional differences in providers’ and patients’ willingness to attend in-person care during that time. Thus the overall results are now being influenced by data from an atypical year. The authors instead might describe usage in 2020 separately, if desired, or revert to the previous time period.

I also do not understand how the number of early adopters could change by ~300 given that that category was defined by adoption in either 2013 or 2014 and therefore should not have been influenced by the addition of 2020 data (except for by the re-categorization of the 32 people originally classified as “always users”). Was there an error in the original analysis? Similarly why did the number of late adopters increase by 1000? Were all of those adopters in 2020?

I do think that the manuscript would benefit from the addition of the provider usage frequency information to the supplement as a complement to the data presented in the primary manuscript.

Please remove statements such as “indicating that the increase in the prevalence of these chronic conditions in the populations that the providers served did slightly impact whether or not they provided IBT obesity services to their patients overall” and any other causal statements that are based on descriptive differences that did not achieve statistical significance (and could not determine causality to begin with). Note that some of these ORs were actually significant in the opposite direction (% pts with diabetes was associated with 14% lower odds of adoption).

Odds ratios for region are only interpretable in comparison to the reference category (Midwest).

The phrasing of this statement is confusing, please reword (and I would recommend changing in favor of a description of the results since the ORs are listed in the table): “Compared with never users, for every 10 percentage points increase in patients with hypertension, diabetes, and hyperlipidemia among later adopters, the ORs were 0.94 (95% CI 0.90, 0.98), 0.92 (0.90, 0.95), and 1.26 (1.22, 1.30), respectively.”

For Table 2 what was the percentage of missing data for the marked outcomes?

7. PLOS authors have the option to publish the peer review history of their article (what does this mean?). If published, this will include your full peer review and any attached files.

Reviewer #1: **Yes: **Hanna Gyllensten

Reviewer #2: **Yes: **Jena S Tronieri

---

## [Decision Letter · Decision Letter 2]

2 Mar 2023

Primary Care Provider Uptake of Intensive Behavioral Therapy for Obesity in Medicare Patients, 2013-2019

PONE-D-22-07772R2

Dear Dr. Luo,

We’re pleased to inform you that your manuscript has been judged scientifically suitable for publication and will be formally accepted for publication once it meets all outstanding technical requirements.

Kind regards,

Juan A López-Rodríguez

Academic Editor

PLOS ONE

Additional Editor Comments (optional):

Reviewers' comments:

Reviewer's Responses to Questions

**Comments to the Author**

1. If the authors have adequately addressed your comments raised in a previous round of review and you feel that this manuscript is now acceptable for publication, you may indicate that here to bypass the “Comments to the Author” section, enter your conflict of interest statement in the “Confidential to Editor” section, and submit your "Accept" recommendation.

Reviewer #1: All comments have been addressed

Reviewer #2: All comments have been addressed

2. Is the manuscript technically sound, and do the data support the conclusions?

Reviewer #1: Yes

Reviewer #2: (No Response)

3. Has the statistical analysis been performed appropriately and rigorously? 

Reviewer #1: Yes

Reviewer #2: (No Response)

4. Have the authors made all data underlying the findings in their manuscript fully available?

Reviewer #1: Yes

Reviewer #2: (No Response)

5. Is the manuscript presented in an intelligible fashion and written in standard English?

Reviewer #1: Yes

Reviewer #2: (No Response)

6. Review Comments to the Author

Reviewer #1: Thanks for the update, I think the manuscript benefited from the recent revisions. No further comments.

Reviewer #2: I thank the authors for their careful revisions and inclusion of the supplemental tables. The authors have addressed all previous concerns and I have no further comments.

7. PLOS authors have the option to publish the peer review history of their article (what does this mean?). If published, this will include your full peer review and any attached files.

Reviewer #1: **Yes: **Hanna Gyllensten

Reviewer #2: **Yes: **Jena S. Tronieri

---

## [Editor Report · Acceptance letter]

14 Mar 2023

PONE-D-22-07772R2 

Primary care provider uptake of intensive behavioral therapy for obesity in Medicare patients, 2013-2019 

Dear Dr. Luo:

I'm pleased to inform you that your manuscript has been deemed suitable for publication in PLOS ONE. Congratulations! Your manuscript is now with our production department. 

Kind regards, 

on behalf of

Dr. Juan A López-Rodríguez 

Academic Editor

PLOS ONE